# OpenReview forum: "SPA-VL: A Comprehensive Safety Preference Alignment Dataset for Vision Language Model"
_ICLR.cc/2025/Conference — ICLR 2025 Conference Withdrawn Submission_

### Official Review · Reviewer_Ksmk · 2024-11-03

**Soundness:** 3
**Presentation:** 3
**Contribution:** 3
**Rating:** 8
**Confidence:** 4

**Summary:**

This paper contributes a large-scale Safety Preference Alignment dataset for Vision Language Models (VLM), which covers many advantages, e.g., samples, diverse categories, train-friendly, model fit-friendly, do not affect general capability, etc.

**Strengths:**

(1) It contributes a important and large-scale Safety Preference Alignment dataset for the VLM research community
(2) Very rich experiments have been carried out to illustrate the validity of the proposed dataset.

**Weaknesses:**

In Table 12, most of the indicators are on a downward trend.

**Questions:**

Why not evaluate the impact of the dataset used on the performance of MMVet and intuitively feel that the dataset built in this paper will increased its performance.

---

### Official Review · Reviewer_nj4G · 2024-11-04

**Soundness:** 2
**Presentation:** 2
**Contribution:** 3
**Rating:** 3
**Confidence:** 4

**Summary:**

This paper presents a comprehensive safety preference alignment dataset for LVLMs. The dataset contains 6 harmlessness domains and 13 categories involving 10w samples. The construction process is automated and the experimental results show significant improvements in terms of harmlessness and usefulness.

**Strengths:**

1. This paper is well-written and organized. The overall architecture is reasonable and easy to follow the author's statement.
2. The appendix is clear and detailed. This thoroughness ensures that readers can easily understand the methodologies and results discussed in the paper.

**Weaknesses:**

1. The experimental results only present the performance of the proposed dataset, without a broader comparison.
2. The analysis in the introduction lacks depth and evidence. For example, the author can use a specific sample to demonstrate the alignment advantages of this dataset for visual modalities.
3. It seems that there is no discussion on whether there is a data breach issue with this dataset, which may affect the evaluation results.

**Questions:**

1. To verify the effectiveness further, please investigate the impact of the model size on the proposed dataset.
2. Please provide more details for “the alignment of the visual encoder is relatively weak” in the introduction.
3. In the introduction, the authors states that visual encoders need to be aligned. But the improvement still mainly depends on the text of the dataset rather than the images.

---

### Official Review · Reviewer_r42w · 2024-11-04

**Soundness:** 3
**Presentation:** 2
**Contribution:** 2
**Rating:** 5
**Confidence:** 3

**Summary:**

This paper proposes “SPA-VL”, the first dataset for safety preference alignment for VLMs, designed for RLHF training (or DPO). The dataset is composed of 100k quadruples (image, question, chosen response, rejected response) and covers 53 “harm categories”. The dataset construction is automated by mining images from LAION-5B, with CLIP, and by generating questions with (jailbreaked) Gemini, and answers with various VLMs. The preference is collected by the MD-Judge model.

The paper then finetunes models (e.g., LLaVA-1.5, QwenVL) on the SPA-VL dataset (with DPO or PPO) and shows improved safety scores on various benchmarks with limited general performance degradation.

**Strengths:**

* The release of a public dataset for VLM alignment is sound and an important asset for the community, as only aligning the LLM is not enough to ensure full alignment, as the authors rightfully explain (L-40).

* The size and coverage of the dataset are large

* Experiments are extensive to measure the impact the dataset can have on the alignment of multiple VLMs, with ablation on the dataset size, variations in the alignment methodologies (DPO, PPO, projections, etc…).

**Weaknesses:**

* The dataset construction process requires the assessment of unsafe content from the generated questions and answers, which is done with MD-Judge (L-130-132). However, this model is trained to assess natural language QA, without considerations of images. Then, how is the dataset creation specifically tackling *VLM* alignment?

* The dataset construction relies on closed-source, proprietary models (Gemini), whose API may change through time, and hinders the reproducibility of the dataset construction.

* The authors claim that “dataset construction process is fully automated” (L.21, L.69), however, it is later explained that in some cases, questions are manually created (L.183). The claim should be adjusted.

* After a thorough read, it remains unclear to the reviewer what a “(hard) statement” is.

* The safety alignment leads to substantial drop in helpfulness (Table 12). Also the use of color is confusing, as in all cases the alignment worsen all metrics.

* “For images where Gemini could not generate descriptions, we retain the original LAION captions to maintain data diversity.” (L.173) → What are those images? I anticipate a potential bias in the dataset construction, as the caption collection for these images will differ and as it will correlate with what made Gemini unable to generate captions for these images. Can the authors comment on that?

* Are the scores of Table 2 obtained with 30k? 90k? It would be helpful to know for cross reference with Table 12.

* Figure 3: Please use a consistent scale for x-axis (e.g., log). It is confusing in the current way of presenting the table.

* Why is the test set so small compared to the training and validation set? (2x 265 vs, 7k for val and 93k for train).

* The paper will benefit from a thorough proofreading. There are many typos, syntax errors (see below a non exhaustive list).
    * “The LAION-5B dataset is trained with a CLIP model” → the dataset is not “trained”
    * “Responses are classified using MD-Judge to ensuring and for each question” L.198
    * Both spellings “GPT4-V” and “GPT-4V” are used
    * “annoate”
    * Please clean the bibliographic references. In several cases, the year is missing. Also, in several cases, preprint (arxiv) paper versions are cited while the works are under proceedings.
    * There is a space “ “ before parentheses

Further suggestions:
It would be beneficial to define “safety” in this context
It would be beneficial to stress more the specificities of alignment for VLM (compared to LLMs)

**Questions:**

Beyond questions mentioned in the weaknesses, the reviewer has the following question:

* The dataset creation is largely automated. While the reviewer understands the value of it, the construction relies on already aligned (or unaligned by jailbreaking) models. It may introduce biases and data contamination. Can the authors comment on that?

---

### Note · Authors · 2024-11-14

I have read and agree with the venue's withdrawal policy on behalf of myself and my co-authors.